# Mechanisms of Pannexin 1 (PANX1) Channel Mechanosensitivity and Its Pathological Roles

**DOI:** 10.3390/ijms23031523

**Published:** 2022-01-28

**Authors:** Kai Yang, Zhupeng Xiao, Xueai He, Ruotong Weng, Xinyue Zhao, Taolei Sun

**Affiliations:** 1School of Chemistry, Chemical Engineering and Life Science, Wuhan University of Technology, Wuhan 430070, China; 247312@whut.edu.cn (Z.X.); 291707@whut.edu.cn (X.H.); Weng_rt@163.com (R.W.); xyz1667@163.com (X.Z.); 2State Key Laboratory of Advanced Technology for Materials Synthesis and Processing, Wuhan University of Technology, Wuhan 430070, China

**Keywords:** pannexin, mechanotransduction, force-from-lipids model, force-from-filaments model, glaucoma, cancer

## Abstract

Pannexins (PANX) were cloned based on their sequence homology to innexins (Inx), invertebrate gap junction proteins. Although there is no sequence homology between PANX and connexins (Cx), these proteins exhibit similar configurations. The PANX family has three members, PANX1, PANX2 and PANX3. Among them, PANX1 has been the most extensively studied. The PANX1 channels are activated by many factors, including high extracellular K^+^ ([K^+^]_e_), high intracellular Ca^2+^ ([Ca^2+^]_i_), Src family kinase (SFK)-mediated phosphorylation, caspase cleavage and mechanical stimuli. However, the mechanisms mediating this mechanosensitivity of PANX1 remain unknown. Both force-from-lipids and force-from-filaments models are proposed to explain the gating mechanisms of PANX1 channel mechanosensitivity. Finally, both the physiological and pathological roles of mechanosensitive PANX1 are discussed.

## 1. An Introduction to Pannexins

Pannexins (PANX) were cloned based on their sequence homology to innexins (Inx), invertebrate gap junction proteins [1]. It was initially proposed that PANX share many functional features with connexins (Cx), vertebrate gap junction proteins [1]. Although there is no sequence homology between PANX and Cx, these proteins exhibit similar configurations: they all have four α-helical transmembrane (TM) domains, two extracellular loops (EL), one intracellular loop (IL), one intracellular N terminal (NT) and one intracellular C terminal (CT) [2]. The PANX family has three members, PANX1, PANX2 and PANX3. Among them, PANX1 and PANX3 are more similar to each other, while PANX2 has a long CT domain that affects its regulation, targeting and macromolecular interactions [2].

PANX1 mRNA and protein are widely expressed in many tissues, including the eye, liver, kidney and central nervous system (CNS) [3]. Northern blot analysis shows that PANX2 mRNA is highly enriched in the CNS. In addition, PANX2 protein is mainly located in cytoplasmic compartments [4]. PANX3 mRNA and protein are only found in skin and osteoblasts [3,5,6]. The expression of PANX1 and PANX2 is inversely regulated in the brain during development. PANX1 mRNA expression is highest in the embryo and decreases significantly in the adult, while the expression of PANX2 mRNA is low in the prenatal brain but increases considerably during adulthood [7]. Six recent studies using cryo-EM have revealed that the PANX1 channel forms a heptamer [8,9,10,11,12,13]. The structures of PANX2 and PANX3 have not yet been solved. PANX2 is proposed to form an octomer [14], while the oligomerization status of PANX3 remains unstudied.

## 2. Mechanisms of PANX1 Activation

It has been shown that different modes of activation can induce PANX1 to form a pore with either high conductance or low conductance [15,16]. The pore configuration with low conductance occurs in response to voltage-dependent stimuli; it has a unitary conductance of 40–70 pS. Its current is mainly mediated by the movement of Cl^−^ ions [15,16]. The pore configuration with high conductance is formed in the presence of a high extracellular K^+^ concentration ([K^+^]_e_)), high intracellular Ca^2+^ concentration ([Ca^2+^]_i_) and/or mechanical stress. Its unitary conductance is 400–500 pS. This configuration is usually associated with ATP release [15,16]. Although the cryo-EM structure of the PANX1 channel has been solved in several recent studies, it is only consistent with its Cl^−^-selective conformation. The narrowest width of the pore is around 9 Å, which only allows the flux of Cl^−^ [8,9,10,11,12]. In order to act as an ATP release channel, the channel pore of PANX1 should be larger than the diameter of an effectively hydrated ATP molecule, which is around 12 Å [17]; a 9 Å pore is too small for ATP to pass. It is possible that an additional conformation of PANX1 is not captured in these cryo-EM studies. The specific activation mechanisms of PANX1 can be broadly categorized as:[K^+^]_e_: It has been found that the effect of [K^+^]_e_ on PANX1 is dose-dependent. A [K^+^]_e_ larger than 10 mM induces PANX1 channel opening and a [K^+^]_e_ of 130 mM activates the PANX1 channel over a wide range of voltages [18]. The mechanism by which high [K^+^]_e_ induces the opening of PANX1 channels remains unknown. PANX1 current is still activated even when voltage is clamped [18]. Instead, it is proposed that high K^+^-induced activation of PANX1 requires a direct interaction of K^+^ with its first EL [19]. Mutations of amino acids (R75A, S82A, L94A) in the ELs of PANX1 change these responses [20]. However, recent results question the ability of high [K^+^]_e_ to activate PANX1; it has been found that the activation of PANX1 by K^+^ is not observed in all cell types [15,21].[Ca^2+^]_i_: PANX1 channels can be stimulated by Gαq-containing G protein-coupled receptors (GPCRs), including P2Y purinergic receptors [22] and ion channels [23,24,25]. This kind of channel activation is mediated by an increase in [Ca^2+^]_i_. For example, fluid shear stress exerted by flowing blood induces the activation of Piezo1, which increases ATP release and NO production in endothelial cells. These effects are mediated in part by PANX channels activated by [Ca^2+^]_i_ [23]. In addition, the alveolar epithelium in the lung comprises alveolar epithelial type I (ATI) and surfactant secreting type II (ATII) cells. In ATI cells, when mechanical tension is imposed upon the membrane, it triggers the activation of Piezo1 channels in the caveolae. The resulting Ca^2+^ influx leads to the opening of PANX1, which induces ATP release and stimulates the secretion of surfactants from ATII cells [24]. Recently, it has been proposed that a Piezo1–PANX1 complex mediates the stretch-induced ATP release in cholangiocytes. The Piezo1 channel senses the membrane stretch and increases [Ca^2+^]_i_, which activates PANX1 and releases ATP [25]. The mechanism of [Ca^2+^]_i_-induced PANX1 activation has been revealed by a recent study. It proposes that the increase in [Ca^2+^]_i_ induces the activation of Ca^2+^/calmodulin-dependent protein kinase II (CaMKII), which phosphorylates the amino acid residue S394 of PANX1, resulting in its opening and ATP release [26].Src family kinase (SFK)-mediated phosphorylation: In addition to being activated by multiple metabotropic receptors, PANX1 channels can also be opened by ionotropic receptors and chemokine receptors, including N-methyl-D-aspartate receptor (NMDAR) and tumor necrosis factor alpha (TNFα) [27,28,29], which is mediated by the SFK phosphorylation of PANX1 channels. In hippocampal neurons, anoxia induces the opening of PANX1 channels via NMDARs [30,31]. This anoxia-induced PANX1 current is reduced by D-APV (an NMDAR antagonist) as well as by PP2 (an SFK inhibitor). However, MK-801 (an NMDAR pore blocker) does not block PANX1 current, suggesting that ion permeation via NMDARs is not required for this mechanism of activation of PANX1 channels. Instead, NMDARs activate the PANX1 channel metabotropically via SFKs [31,32]. A possible Src phosphorylation site on PANX1 channels is located at Tyr308. Activation of NMDARs increases the phosphorylation of Tyr308 in the presence of PP2 or a PANX1 mutant, which cannot be phosphorylated at Tyr308; the NMDAR-dependent increase at Tyr308 is abolished in cells [32]. SFKs are also involved in the activation of PANX1 channels induced by TNFα in human umbilical vein endothelial cells (HUVECs). TNFα-induced PANX1 opening requires the activation of type-1 TNF receptors and downstream signaling pathways through SFKs [29]. Further, the application of TNFα increases the phosphorylation of Tyr198 in PANX1 channels, and this increase is inhibited by PP2 [29], suggesting that the Tyr198 of PANX1 is a target of SFK phosphorylation in response to TNFα stimulation.Caspase cleavage: At basal conditions, the C-terminus of PANX1 interacts with the channel pore and prevents channel activation. During apoptosis, after caspase 3/7 is activated, it cleaves the C-terminus of PANX1, resulting in constitutive activation of the channel [33]. Even without the occurrence of apoptosis, the direct application of constitutively activated caspase 3/7 also potentiates PANX1 channels, suggesting that the cleavage directly regulates channel opening without the involvement of other apoptotic mediators [34]. In addition, the isolated CT tail of PANX1 blocks the channel pore [34]. Caspase 11 also activates the PANX1 channel by cleavage, which is involved in lipopolysaccharide (LPS)-induced pyroptosis in bone marrow-derived macrophages (BMDMs) [35].Mechanical stimulation: PANX1 channels can also be stimulated by a wide range of mechanical stresses. In *Xenopus* oocytes injected with the cDNA of human PANX1, the PANX1 channel is activated mechanically by suction applied to a patch pipette. The increased channel activity occurs over a wide range of holding potentials when stressed mechanically [36]. The authors also showed that the activation of PANX1 can release ATP [36]. Subsequently, this mechanical sensitivity of PANX1 has been identified in other cells, including erythrocytes, lung epithelium and neurons [37,38,39,40]. Further, PANX1 is activated when cancer cells are subjected to deformation as they travel along the microvasculature, which contributes to cancer metastasis [41]. Moreover, focused ultrasound (FUS) can stimulate ER-localized mechanosensitive PANX1 and result in the release of Ca^2+^ from the ER in invasive cancer cells [42].

## 3. Gating Mechanisms of PANX1 Channel Mechanosensitivity

There are at least two independent gating mechanisms for PANX1 channels. One is mediated by its CT region; this gating mechanism is well studied. The CT region blocks the main pore of PANX1 via a ball and chain mechanism. It acts as a pore plug that can be moved away or cleaved, resulting in opening of the channel [34]. Consistently, the comparison between cryo-EM images of native PANX1 and CT-deleted PANX1 reveals that the conformation of the channel is not affected by cleavage of the C-terminus. In addition, progressive deletion of the CT region of PANX1 channels causes their stepwise activation, along with graded changes in single-channel conductance and ATP/dye permeation of PANX1 [21]. A second gating mechanism acts through the side tunnels of PANX1. PANX1 protein is a heptamer; side channels are formed by the shallow crevice between the upper intracellular domains (ICDs) of protomers. They have connections to the main pore and are gated by a linker between the NT helix and the first TM helix [13], but the structural mechanism for PANX1 mechanosensitivity has not been revealed by recent cryo-EM studies [8,9,10,11,12,13].

Along with structural and biophysical studies of disparate families of mechanosensitive ion channels, our understanding of PANX mechanosensitivity is developing rapidly. Two classic models for the mechanosensitive gating of PANX have been proposed, the force-from-lipids model and the force-from-filaments model. Recently, a third model has been suggested; in this hybrid model, the mechanotransduction of PANX is induced by both lipids and the cytoskeleton.

### 3.1. Force-from-Lipids Model

The force-from-lipids model proposes that the channel is opened by the force transmitted through the lipid bilayer directly, without the involvement of the cytoskeleton or extracellular matrix [43,44]. After the ion channel is purified and reconstituted in a lipid bilayer, if it can be mechanically activated, then it can be considered “inherently mechanosensitive”. In vitro liposome reconstitution has become a gold standard for identifying inherently mechanosensitive ion channels [45]. This model of PANX activation is supported by the PANX1 channel in excised membrane patches being activated by negative pressure (acting through a patch pipette by suction) [36,37].

The core mechanical force for gating mechanosensitive ion channels is generated by a change in the transbilayer pressure profile of the lipid bilayer. The anisotropy of the transbilayer pressure profile is determined by three factors. One is the repulsion between the hydrophilic lipid heads. The second is the steric repulsion among the lipid tails, which depends on the degree of lipid saturation. The third is the attraction between phospholipid molecules at the water–lipid interface, caused by the hydrophobicity of the phospholipid tails. This force prevents water molecules from entering the lipid bilayer [46]. In an idealized lipid bilayer with two identical monolayers, the transbilayer pressure profile is usually symmetric. There are negative peaks at the water–lipid interface and repulsive positive peaks in both the head and tail regions. The presence of a mechanosensitive ion channel will induce the pressure profile to become asymmetric [47]. Reciprocally, an asymmetric pressure profile could affect the activity of mechanosensitive ion channels.

The change in asymmetric pressure profile can be caused by either hydrophobic mismatch or bilayer curvature. Hydrophobic mismatch is induced by stretching a bilayer [48]. When the bilayer stretches, the membrane thins and causes hydrophobic mismatch between the membrane-facing domains of ion channels and the bilayer, which opens ion channels (Figure 1A) [46]. Bilayer curvature is generated by the asymmetric insertion of amphipaths into lipid bilayers (Figure 1A) [43,49]. The activity of many mechanosensitive channels, including Piezo1 and small-conductance mechanosensitive channel (MscS), can be modulated by these amphipathic molecules [46].

The transbilayer pressure profile can be determined using either computational simulations [50] or experimental measurement [51]. When lipids with a different degree of unsaturation are incorporated into the phospholipids of cells, they have the ability to modulate the transbilayer pressure profile, which influences the ability of mechanical stress to activate mechanosensitive ion channels in the membrane. For example, the calculated value of the intra-bilayer pressure in the lipid bilayer is 250 atm for monounsaturated (18:1) lipids, while it is 350 atm for polyunsaturated (18:2 and 18:3) lipids [47,50]. These values have also been confirmed by NMR spectroscopy [51]. Consistently, depending on their chain length and degree of unsaturation, fatty acids have the ability to regulate PANX1 gating differentially. Acute application of saturated fatty acids (SFAs), such as palmitic acid (PA) (16:0) and stearic acid (SA) (18:0), in human and rat liver cell lines significantly potentiates PANX1 channel activity [52]. PA can also open the PANX1 channel in human renal tubule epithelial cells (HK-2 cells) [53]. In addition, upon exposure to palmitate, in macrophages, PANX1 is opened and releases nucleotides to attract neutrophils [54]. PANX3 has the same role in L6 myotubes [55]. The effect of monounsaturated FAs on the PANX1 channel is quite complex. In human and rat liver cell lines, neither palmitoleic acid (PO) (16:1) nor oleic acid (OA) (18:1) modulates PANX1 channel activity [52]. However, in bovine polymorphonuclear leukocyte cells (PMN), OA induces PANX1 opening, resulting in the formation of neutrophil extracellular traps (NETs) and the release of extracellular ATP [56]. Polyunsaturated FAs can either reduce or potentiate PANX1 channel activity. In *Xenopus* oocytes, arachidonic acid (AA) (20:4) reduces the activity of PANX1 channels [57]. However, in PMN, linoleic acid (LA) (18:3) induces NET formation and extracellular ATP release via the opening of PANX1 [56]. To summarize, although PANX1 mechanosensitivity is greatly influenced by the membrane lipid composition, whether it acts through a change in the membrane transbilayer pressure profile remains unstudied.

### 3.2. Force-from-Filaments Model

The alternative mechanism is that the PANX channel is gated via the cytoskeleton or extracellular matrix, which interacts with the PANX channel (Figure 1B). A study using co-immunoprecipitation and co-sedimentation assays has shown that PANX1 interacts with actin through its C-terminus [58]. Using liquid chromatography and tandem mass spectrometry (LC/MS), both actin and actin-related protein 3 (Arp3) have been identified to bind PANX1 directly [59]. A recent study has shown that, when PANX1 is overexpressed in rhabdomyosarcoma (RMS), both RNA sequencing and co-immunoprecipitation coupled to high performance liquid chromatography/electrospray ionization tandem mass spectrometry (HPLC/ES-MS) reveal that PANX1 interacts with many cytoskeleton-associated proteins physically [60].

Consistently, the inhibition of Rho kinase, Ras homolog family member A (RhoA) or myosin light chain (MLC) kinase, which disrupts the actin cytoskeleton, significantly reduces ATP release via mechanosensitive PANX1 channels [38]. Further, the treatment of Schwann cells using a Rho GTPase inhibitor or small interfering RNA (siRNA) targeting Rho or cytochalasin D results in a decrease in hypotonicity-induced ATP release via PANX1 [61]. In addition, PANX1 is associated with collapsing response mediator protein 2 (Crmp2), which is a well-known microtubule-stabilizing protein [62].

Not surprisingly, PANX1 activity in turn triggers a great deal of cellular processes involving the cytoskeleton. In C6 glioma cells, the expression of PANX1 controls the actomyosin system and accelerates the assembly of multicellular C6 glioma aggregates [63]. This effect is mediated by the release of ATP from the PANX1 channel as well as activation of the P_2_X_7_ receptor [63]. Another example occurs in dendritic cells (DCs); when ATP is released from injured cells, it acts as a danger signal to attract DCs to the site of injury. During this process, the actin cytoskeleton of DCs is reorganized, which is PANX1 channel- and P_2_X_7_ receptor-dependent [64]. Further, in human skin fibroblasts, the inhibition of PANX1 enhances actin dynamics and cell motility [65]. There is a feedback mechanism between PANX1 and the cytoskeleton. PANX1 regulates cytoskeletal dynamics, and the cytoskeleton in turn influences PANX1 activation.

### 3.3. Hybrid Model

In fact, although the PANX1 channel can be activated by negative pressure in excised membrane patches, the cytoskeleton is still tightly associated with these membrane patches [66], so more work has to be done to confirm the inherent mechanosensitivity of the PANX1 channel. In addition, in order to comply with the force-from-lipids model, a mechanosensitive ion channel should have a domain that acts as a sensor to detect its interaction with the lipid bilayer; this domain should also connect directly to the channel pore [46,67]. Whether the PANX1 channel has this domain remains unknown.

It turns out that these two models of mechanosensitive PANX channels are not mutually exclusive. For example, no mechanoreceptor potential C (NOMPC) contains 29 ankyrin repeats, which are required for its activation by force [68]. It is also sensitive to the force-from-lipids paradigm. A residue of NOMPC that interacts with lipids is required for mechanically induced channel activation [69]. Conversely, the cytoskeleton and extracellular matrix also affect inherently mechanosensitive channels. TREK-1 is intrinsically mechanosensitive, but its activity can also be modulated by cytoskeletal elements [70]. It is proposed that PANX1 gating follows the force-from-lipids model, but that the extracellular matrix and cytoskeleton can significantly affect the forces applied to PANX1.

## 4. Mechanosensitive PANX1 in Physiological Processes

### 4.1. Airway Defense

In the lung, the primary innate defense is mediated by the mucociliary clearance process, which removes foreign pathogens from the airway. Nucleotides and nucleosides act on purinergic receptors in the epithelial surface to regulate the key components of mucociliary clearance. Mechanical stress in the lung mainly comes from tidal breathing, coughing and cell swelling during hypotonic gland secretions; they can strongly stimulate ATP release in the airways via PANX1 channels. In lung epithelial cells, PANX1 is activated by hypotonically induced cell swelling [38] or hydrostatic pressure [40]. In addition, ATP release from hypotonically swollen human bronchial epithelial (HBE) cells is partially inhibited by PANX1 blockers or by knocking down the expression of PANX1 [71].

### 4.2. Ca^2+^ Wave Signaling

There is a positive feedback mechanism involving mechanosensitive PANX1 to mediate the initiation and propagation of intercellular Ca^2+^ waves. In a single layer of primary corneal endothelial cells, a Ca^2+^ wave is elicited by a mechanical stimulus; it propagates to the neighboring cells rapidly. PANX1 channels are implicated in this initiation and propagation of Ca^2+^ waves [72]. It is proposed that, when PANX1 is activated, it releases ATP from the cell. The increase in extracellular ATP activates P2Y purinergic receptors on cells within diffusion distance; P2Y receptors then stimulate phospholipase C (PLC), allowing an increase in inositol 1,4,5-triphosphate (IP_3_), which induces the release of Ca^2+^ from intracellular stores. The increase in [Ca^2+^]_i_ results in the opening of PANX1 channels and ensuing ATP release, providing a new source of ATP diffusing to cells further away (Figure 2) [22].

## 5. Mechanosensitive PANX1 in Diseases

### 5.1. Glaucoma

Glaucoma is caused by elevated intraocular pressure (IOP) in the eye and has been associated with the death of retinal ganglion cells (RGCs). Both increased production and decreased outflow of aqueous humor within the anterior chamber result in increased IOP [73]. Positive correlations between IOP levels and RGC loss have been reported in glaucomatous rodents [74].

Recent studies have shown that PANX1 is involved in glaucoma pathogenesis [39,75,76]. Using ex vivo bovine eyecup preparations, an increase in pressure across the retina triggers ATP release, which can be blocked by PANX1 inhibitors [75]. A subsequent study suggested that RGCs themselves can mediate ATP release via mechanosensitive PANX1. In response to the stretching or swelling induced by hypotonic solution, RGCs can release ATP via PANX1 channels since this release is inhibited by carbenoxolone (Cbx), probenecid (Pbn) and ^10^PANX. In addition, swelling-induced current is inhibited by apyrase, A438079, Cbx and Pbn. ATP release via the PANX1 channel is also required for the regulatory volume decrease (RVD) of RGCs. It is proposed that excessive ATP is released when mechanosensitive PANX1 is open, which leads to the activation of purinergic receptors and induces RGC death [39].

In addition, astrocytes isolated from optic nerve heads also increase ATP release under mechanical stress; PANX1 is involved since both Cbx and Pbn significantly reduce ATP release. Optic nerve head astrocytes isolated from PANX1 −/− mice show deficits in both baseline and swelling-induced ATP release. RVD of astrocytes requires both PANX1 activation and the presence of extracellular ATP. Further, the expression of PANX channels is increased upon chronic stretching in isolated astrocytes and the Tg-MYOC^Y437H^ mouse model of chronic glaucoma [76].

### 5.2. Cancer

During metastasis, primary tumor cells enter the blood stream from their original site and are distributed throughout the whole body via the blood vessels. Around 90% of tumor cells die during this process due to mechanical stress acting upon them in blood vessels. Despite this hindrance, a few of these cancer cells still survive and form lethal metastatic colonies in organs.

Using a whole-transcriptomic RNA sequencing (RNA-seq) technique, Furlow et al. identified a truncated form of the PANX1 channel, PANX1^1–89^, in highly metastatic human breast cancer cells. When this truncated form of PANX1 is co-expressed with wildtype full-length PANX1 channels, ATP release from the cells is significantly increased [41]. The ATP released under mechanical stress is required for the survival of metastatic cancer cells via the activation of purinergic receptors. Consistently, when the PANX1 channel is inhibited in cancer cells, these cells still survive under hypotonic stress in the presence of extracellular ATP [41]. Moreover, pharmacological inhibition of PANX1 reduces breast cancer metastasis in vivo [41]. Therefore, PANX1 channels may prove to be a therapeutic target for cancer metastasis. However, this paper does not provide a direct link between mechanosensitive PANX1 channels and cancer metastasis; more work has to be done to confirm it.

In addition, previous studies have showed that FUS stimulation has the ability to induce oscillatory Ca^2+^ dynamics in many invasive cell lines [77]. In invasive PC-3 cells, FUS directly activates mechanosensory PANX1 localized in the ER and results in the release of Ca^2+^ from the ER [42]. Ultrasound can penetrate the plasma membrane and activate the PANX1 channel in the ER membrane directly. It exerts acoustic radiation force on the membrane by increasing membrane tension or deforming the lipid bilayer [78]. Additionally, FUS stimulation induces the release of cytokines/chemokines from invasive cancer cells, indicating that FUS could be used to improve cancer immunotherapy [42].

## 6. Conclusions

Although there is a great deal of evidence to support the mechanosensitivity of the PANX1 channels, the involvement of other channels cannot be excluded. For example, leucine-rich repeat-containing protein 8 (LRRC8, also called SWELL1) channels, which are a major component of the volume regulated anion channel (VRAC), have many properties similar to those of PANX1. They are also sensitive to Cbx and have the ability to release ATP [79].

Furthermore, although a considerable amount of work has linked the mechanosensitivity of PANX1 to several diseases, how PANX1 channels are sensitive to the mechanical stimuli remains unknown. Furthermore, although the structure of PANX1 has been determined using cryo-EM, it does not explain the mechanism of its mechanosensitivity. It is proposed that PANX1 might have a domain that interacts with the lipid bilayer and senses the mechanical force, but further studies are required to identify it.

In addition, there are no pharmacological tools available that can be used to dissect the role of mechanosensitive PANX1. It is important to design tools to specifically block the mechanosensitive responses of the PANX1 channel, which could help us to understand the role of the mechanosensitive responses of PANX1 in many physiological processes and pathological diseases. These studies could potentially provide a therapeutic target for many diseases where the mechanosensitive activation of PANX1 channels is involved.

## Figures and Tables

**Figure 1 ijms-23-01523-f001:**
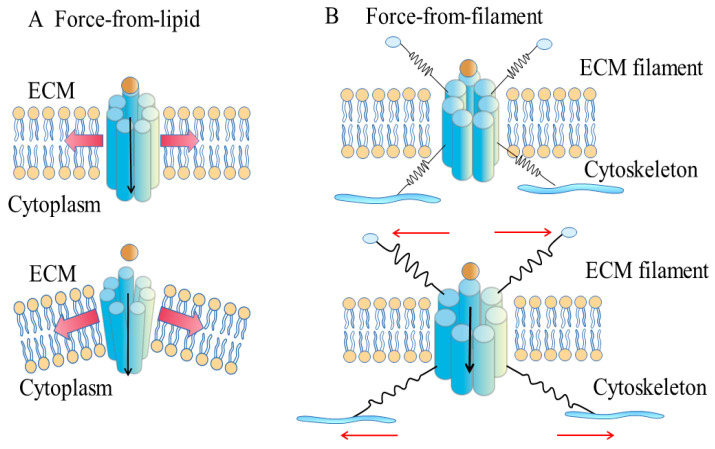
Two gating models of the mechanosensitive PANX1 channel. A. Force-from-lipids model. The PANX1 channel is opened by the force transmitted through the lipid bilayer directly, without the involvement of the cytoskeleton or extracellular matrix. The core mechanical force for gating PANX1 is generated by the change in the transbilayer pressure profile of the lipid bilayer, which includes both hydrophobic mismatch and bilayer curvature. Hydrophobic mismatch can be induced by stretching a bilayer, while bilayer curvature is generated by the asymmetric insertion of amphipaths into lipid bilayers. B. Force-from-filaments model. The PANX1 channel is gated via the cytoskeleton or extracellular matrix, which interacts with the PANX1 channel.

**Figure 2 ijms-23-01523-f002:**
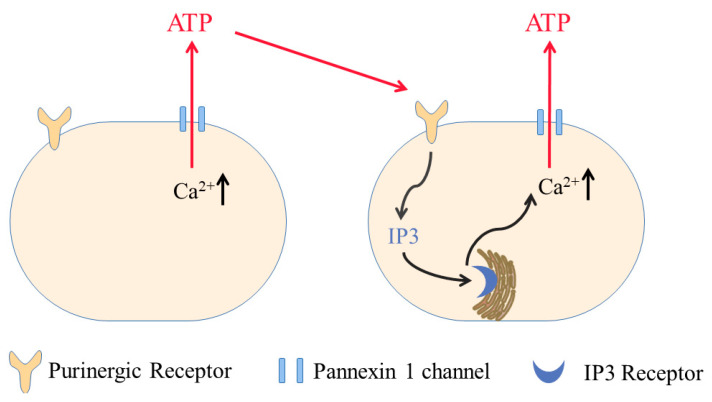
PANX1 is involved in the initiation and propagation of intercellular Ca^2+^ waves. When [Ca^2+^]_i_ increases in one cell, it opens PANX1 channels and releases ATP from the cell. The increase in extracellular ATP activates P2Y purinergic receptors on cells within diffusion distance; P2Y receptors then stimulate phospholipase C (PLC), allowing an increase in inositol 1,4,5-triphosphate (IP_3_), which induces the release of Ca^2+^ from intracellular stores. The increase in [Ca^2+^]_I_ results in the opening of PANX1 channels and ensuing ATP release, providing a new source of ATP diffusing to cells further away.

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
