# Peer review of "Mechanisms of Pannexin 1 (PANX1) Channel Mechanosensitivity and Its Pathological Roles"

_ijms, 2022, doi:10.3390/ijms23031523_

Round 1
Reviewer 1 Report
Major
There are too many grammatical errors. I started to scan, but after the first page I decided that I request the authors to provide a corrected manuscript with acceptable English.
line 12: eliminate . after (Cx)
line 14: “evidence” has no plural form
line 14: it is strange to mention PANX1 without having clarified how many PANXs there are
line 16: remains -> remain
line 16: the phrase “to explain it” is awkward (what is “it”?)
line 24: provide a ref after 1st sentence
line 26: eliminate . after (Cx)
line 30: replace “including” by “comprising”
line 34: replace “kidneys” by “kidney”
line 35: replace “PANX2 proteins are mainly located …” by “PANX2 protein is mainly located …”
line 36: replace “PANX3 mRNAs and proteins are only …” by “PANX3 mRNA and protein are only …”
line 38: replace “The expressions of PANX1 and PANX2 are inversely” by “The expression of PANX1 and PANX2 is inversely”
line 39: replace “PANX1 mRNA expression is the highest” by “PANX1 mRNA expression is highest”
line 40: “Whereas” cannot make up for an isolated sentence.
Author Response
Response to Reviewer1:
Major
1: There are too many grammatical errors. I started to scan, but after the first page I decided that I request the authors to provide a corrected manuscript with acceptable English.
A: Sorry for our mistakes, the revised manuscript has been professionally proofread.
2: line 12: eliminate . after (Cx)
A: It has been corrected, see page 1, line 12.
3: line 14: “evidence” has no plural form
A: I have rephrased this sentence, now this word is deleted.
4: line 14: it is strange to mention PANX1 without having clarified how many PANXs there are
A: Sorry for our mistake, now we add one sentence to mention it. See page 1, line 12-13. “The PANX family has three members, PANX1, PANX2 and PANX3. Among them, PANX1 has been most extensively studied.”
5: line 16: remains -> remain
A: It is corrected, see page 1, line 16.
6: line 16: the phrase “to explain it” is awkward (what is “it”?)
A: The reviewer is right, we rephrased this sentence, see page 1, line 17-18 “Both force-from-lipids and force-from-filaments models are proposed to explain the gating mechanisms of PANX1 channel mechanosensitivity”.
7: line 24: provide a ref after 1st sentence
A: this point is well taken, now we add one reference, see page 1, line 25.
8: line 26: eliminate . after (Cx)
A: this dot is deleted, see page 1, line 26.
9: line 30: replace “including” by “comprising”
A: We rephrased this sentence, see page 1, line 30-31.
10: line 34: replace “kidneys” by “kidney”
A: it has been corrected, see page 1, line 35.
11: line 35: replace “PANX2 proteins are mainly located …” by “PANX2 protein is mainly located …”
line 36: replace “PANX3 mRNAs and proteins are only …” by “PANX3 mRNA and protein are only …”
line 38: replace “The expressions of PANX1 and PANX2 are inversely” by “The expression of PANX1 and PANX2 is inversely”
line 39: replace “PANX1 mRNA expression is the highest” by “PANX1 mRNA expression is highest”
A: Thanks for the reviewer’s suggestions, they are corrected, see page 1, line 36, 37, 38, 39.
15: line 40: “Whereas” cannot make up for an isolated sentence.
A: This point is well taken, see page 1, line 40.
Reviewer 2 Report
The present review summarizes current knowledge on potential mechanisms of Panx mechanosensitivity and the role of Panx-formed channels in various pathological processes. The review is well-written, its main idea is innovative and attracting. However, there are some problems needed to be considered before the manuscript is accepted.
MAJOR
1) Title:
- “mechanosensitive” should probably be changed to “mechanosensitivity”;
- “pannexin” should probably be changed to “pannexin 1” since it is mainly addressed in the review (this also concerns section heading).
2) It is not clear why the authors concentrated on the pathological aspect of the problem and did not pay attention to the participation of pannexin mechanosensitive channels in the normal regulation of cell functions. To fill this gap, section “3. The gating of PANX1 channels” should be placed after section “4. Mechanical stimulation of PANX channels” and expanded by describing the normal function of mechanosensitive pannexin 1 channel (section title should be changed accordingly).
3) Information about the possible interaction of pannexin and other channels is now given only in the conclusion, while this interesting topic can be addressed in a separate section (see, for example, PMID 27797339, 29349673, 33229726 etc.).
MINOR
4) Lines 251-253 (RhoA-dependent control of pannexin 1 channel): Rho-kinase, which is important participant of this mechanism, must be mentioned here (see Fig.6C in Seminario-Vidal et al.)
5) Many abbreviations are introduced in the text without explanation: MscS (line 203), NOMPC (line 280), RGC (line 299) etc. Must be carefully checked and corrected.

Author Response
Response to Reviewer2:
The present review summarizes current knowledge on potential mechanisms of Panx mechanosensitivity and the role of Panx-formed channels in various pathological processes. The review is well-written, its main idea is innovative and attracting. However, there are some problems needed to be considered before the manuscript is accepted.
MAJOR
1. Title “mechanosensitive” should probably be changed to “mechanosensitivity”;
“pannexin” should probably be changed to “pannexin 1” since it is mainly addressed in the review (this also concerns section heading).
A: These points are all well taken, see the title of the manuscript and section heading
2. It is not clear why the authors concentrated on the pathological aspect of the problem and did not pay attention to the participation of pannexin mechanosensitive channels in the normal regulation of cell functions. To fill this gap, section “3. The gating of PANX1 channels” should be placed after section “4. Mechanical stimulation of PANX channels” and expanded by describing the normal function of mechanosensitive pannexin 1 channel (section title should be changed accordingly).
A: Based on the reviewer’s suggestions, we reorganized the manuscript and added one section to describe the normal function of mechanosensitive pannexin 1 channel, see page 7, line 2428-2451.
“ 4. Mechanosensitive PANX1 in Physiological Processes
4.1. Airway Defense
In the lung, the primary innate defense is mediated by the mucociliary clearance process which removes foreign pathogens from the airway. Nucleotides and nucleosides act on purinergic receptors in the epithelial surface to regulate the key components of mucociliary clearance. Mechanical stress in the lung mainly comes from tidal breathing, coughing and cell swelling during hypotonic gland secretions; they can strongly stimulate ATP release in the airways via PANX1 channels. In lung epithelial cells, PANX1 is activated by hypotonically induced cell swelling [38] or hydrostatic pressure [40]. In addition, ATP release from hypotonically swollen human bronchial epithelial (HBE) cells is partially inhibited by PANX1 blockers or by knocking down the expression of PANX1 [71].
4.2. Ca2+ Wave Signaling
There is a positive feedback mechanism involving mechanosensitive PANX1 to mediate the initiation and propagation of intercellular Ca2+ waves. In a single layer of primary corneal endothelial cells, a Ca2+ wave is elicited by a mechanical stimulus; it propagates to the neighboring cells rapidly. PANX1 channels are implicated in this initiation and propagation of Ca2+ waves [72]. It is proposed that when PANX1 is activated, it releases ATP from the cell. The increase of extracellular ATP activates P2Y purinergic receptors on cells within diffusion distance; P2Y receptors then stimulate phospholipase C (PLC), allowing an increase in inositol 1,4,5-triphosphate (IP3) which induces the release of Ca2+ from intracellular stores. The increase of [Ca2+]i results in the opening of PANX1 channels and ensuing ATP release, providing a new source of ATP diffusing to cells further away (Figure 2) [22]”
In addition, we placed section “The gating of PANX1 channels” after we talked “Mechanical stimulation of PANX channels”. See page 3. Line 1036-1061.
3. Information about the possible interaction of pannexin and other channels is now given only in the conclusion, while this interesting topic can be addressed in a separate section (see, for example, PMID 27797339, 29349673, 33229726 etc.).
A: We appreciate the reviewer’s suggestion, now we address the possible interaction of pannexin and other channels in page 2, line 354-369.
“This kind of channel activation is mediated by an increase of [Ca2+]i. For example, fluid shear stress exerted by flowing blood induces the activation of Piezo1, which increases ATP release and NO production in endothelial cells. These effects are mediated in part by PANX channels activated by [Ca2+]i [23]. In addition, the alveolar epithelium in the lung comprises alveolar epithelial type I (ATI) and surfactant secreting type II (ATII) cells. In ATI cells, when mechanical tension is imposed upon the membrane, it triggers the activation of Piezo1 channels in the caveolae. The resulting Ca2+ influx leads to the opening of PANX1, which induces ATP release and stimulates the secretion of surfactants from ATII cells [24]. Recently, it has been proposed that a Piezo1–PANX1 complex mediates the stretch-induced ATP release in cholangiocytes. The Piezo1 channel senses the membrane stretch and increases [Ca2+]i, which activates PANX1 and releases ATP [25].
The mechanism of [Ca2+]i-induced PANX1 activation has been revealed by a recent study. It proposes that the increase of [Ca2+]i induces the activation of Ca2+/calmodulin-dependent protein kinase II (CaMKII), which phosphorylates the amino acid residue S394 of PANX1, resulting in its opening and ATP release [26].”
MINOR
4. Lines 251-253 (RhoA-dependent control of pannexin 1 channel): Rho-kinase, which is important participant of this mechanism, must be mentioned here (see Fig.6C in Seminario-Vidal et al.)
A: We agree with the reviewer and mentioned Rho-kinase in page 6, line 2076-2077. “Consistently, the inhibition of Rho kinase, Ras homolog family member A (RhoA) or myosin light chain (MLC) kinase, which disrupts the actin cytoskeleton”.
5. Many abbreviations are introduced in the text without explanation: MscS (line 203), NOMPC (line 280), RGC (line 299) etc. Must be carefully checked and corrected.
A: Sorry for these mistakes, these abbreviations are all explained when they appear at the first time in the text, see page 4, line 1549; page 6, line 2103; page 8, line 2695.
Round 2
Reviewer 1 Report
The paper has been suitably revised.
This manuscript is a resubmission of an earlier submission. The following is a list of the peer review reports and author responses from that submission.